# Open-Plan Offices: Comparison of Methods for Measuring Psychoacoustic Intelligibility Parameters

**María P. Serrano-Ruiz** [1,*] **, José A. Yarza-Acuna** [2]**, Erwin A. Martinez-Gomez** [1] **and Gabriel Ibarra-Mejía** [3]

1    Departamento de Ingeniería Industrial y Manufactura, Instituto de Ingeniería y Tecnología, Universidad Autónoma de Ciudad Juárez, Ciudad Juárez 32584, Chihuahua, Mexico; emartine@uacj.mx
2    Escuela de Sistemas Unidad Torreón, Universidad Autónoma de Coahuila, Torreón 27087, Coahuila, Mexico; joyarzaa@uadec.edu.mx
3    Department of Public Health Sciences, University of Texas at El Paso, El Paso, TX 79968, USA; gabmejia@utep.edu
*    Correspondence: maria.serrano@uacj.mx

**Abstract:** The acoustic conditions of open-plan office spaces influence the well-being and productivity perceived by users. However, with an inadequate evaluation of the workspace, acoustic design in open-plan offices can be a factor that alters user performance. Such is the case in Mexico, where there are no adequate standards to evaluate specific acoustic conditions such as intelligibility. For this reason, this case study aims to evaluate different types of measurement methods for intelligibility. This study was carried out at a university in northern Mexico. The sound measurements were based on the Mexican standard for noise analysis and the ISO 3382-part 3 standards for acoustic measurements for open-plan offices. The psychoacoustic parameters evaluated were reverberation and intelligibility, using objective methods determined on S/N and subjective methods based on loss of consonant, where it was analyzed the distance between the sound source and zones classified by building design characteristics. The results indicated at which points the intelligibility effects increased. We also observed that reverberation remained stable in this office and that the subjective methods presented a larger measured sound effect than the objective methods. This finding establishes that subjective methods conform to Lognormal behavior, which is applicable to other linguistic elements describing speech behavior.

**Keywords:** open-plan office; intelligibility; reverberation time; Speech Transmission Index; Articulation Loss of Consonants

## 1. Introduction

Open-plan offices have gained popularity as spaces that foster communication among users, primarily due to the favorable acoustic conditions that enhance speech intelligibility [1]. However, despite this advantage, open-plan offices often suffer from high levels of background noise caused by office equipment and multiple speakers. This background noise has been associated with user discomfort, increased stress, and reduced productivity [2]. Consequently, addressing the acoustic challenges of open-plan offices through appropriate architectural design elements becomes crucial. Building materials and furniture influence the propagation of sound waves, affecting factors such as reverberation and speech intelligibility within the office enclosure [3,4].

It is evident that inadequate architectural acoustic design in open-plan offices can result in the clear audibility of background conversations, leading to distraction and negatively impacting workers' cognitive performance and overall well-being [5,6]. The distraction caused by voices arises from the allocation of attention resources when individuals engage in conversation, and the clarity of speech directly correlates with the level of distraction experienced by individuals.

To evaluate the acoustic conditions in open-plan office environments, the parameter of speech intelligibility becomes essential, as it directly influences the comfort and effectiveness of workspaces. The soundscape of open-plan offices, particularly when multiple speakers are present, has been identified as a significant source of distraction, interfering with task concentration, and causing annoyance for workers [7,8]. Numerous studies have linked background noise generated by multiple talkers to cognitive performance deficits in open-plan office users, affecting attention, concentration, memory, collaboration, and task difficulty [9–13].

As a psychoacoustic parameter, intelligence has been widely employed to study the relationship between architectural acoustic design in open-plan offices and users' cognitive performance. It comprises physical and semantic factors. The physical factor involves objective methods that measure the propagation of sound waves and the reverberation of space, representing how speech is transmitted [14–16]. Objective methods, such as those standardized in ISO 3382-3:2012, incorporate parameters based on the signal-to-noise ratio (SNR) and the relationship between reverberation and intelligibility [1,17]. On the other hand, subjective methods utilize standardized assessments of perceived intelligibility quality using categorized scales in specific study areas [18,19].

While objective methods often utilize the Speech Transmission Index (*STI*) to measure irrelevant noise using *SNR*, the relationship between *STI* and specific task performance characteristics, such as collaboration and accuracy, has not shown significant results [20–23]. Additionally, the Reverberation Time (*RT*), which quantifies the decay duration of sound emitted by a source, and the Percentage of Loss of Articulation of Consonants *(%ALCons)*, which predicts intelligibility by measuring the loss of information in unrecognized logatoms, are utilized as parameters in objective and subjective methods, respectively [15].

A psychoacoustic approach in multi-speaker environments, as highlighted by Braat-Eggen [24], has shown the potential to improve the writing performance of open-plan office users. The perceived soundscape in open-plan offices has been analyzed to evaluate its effects on cognitive performance, with physical-environmental factors, such as Activity Based Work (ABW) designs, associated with self-perceived unproductivity and the well-being of workers [25]. In Mexico, the ABW design is widely adopted in various workplaces, both in public and private companies, due to its cost-effectiveness and versatility. This design approach offers open areas for coexistence, shared cubicles to optimize space utilization, and enclosed offices or study areas to provide individuals with increased privacy for their tasks. ABW design has found extensive implementation in Mexican workplaces, including universities, where ABW offices are utilized in libraries as consultation or study areas. However, it is important to note that background noise is often generated despite the intention of creating quiet spaces in ABW design areas. Furthermore, in many cases, the architectural acoustic design in these ABW spaces, especially in Mexico, does not receive sufficient consideration in terms of optimizing the performance and well-being of the users.

While the ABW design offers versatility in workplaces, providing open areas for coexistence and shared cubicles, it is crucial to address the challenges posed by background noise. It is noteworthy that the existing literature on this topic is limited, and there is a need for further research to develop effective solutions for optimizing the acoustic performance of open-plan offices. This study aims to address this research gap by comparing objective and subjective methods for measuring intelligibility and evaluating their behavior and significant effects in an open-plan office with ABW design.

By conducting comprehensive acoustic and statistical analyses, this study seeks to provide insights into the behavior of various parameters, such as *STI, RT*, and *%ALCons*, and their comparative performance as psychoacoustic indicators of intelligibility. The findings of this study will contribute to advancing our understanding of the semantic factor of the intelligibility parameter. Furthermore, this research extends the existing literature by comparing these parameters with each other, highlighting the advantages and unique contributions of the proposed work.

In the subsequent sections, we will present a detailed Methodology, Results and Discussion. Finally, we will conclude the study by summarizing the key findings and their implications for optimizing the acoustic design of open-plan offices.

## 2. Materials and Methods

The present study consists of comparing the methods of measuring intelligibility to evaluate the behavior of the sound conditions and the size of the effect they have on the parameters at the measured points. The study was carried out in the library of a university in northern Mexico, where the computer service area was selected. The computer service area is an open-plan office with an ABW design. The office has a space of 789 m$^2$, five private offices, a boardroom, and a data center area. In the open-plan office, there are 24 shared workstations with four cubicles per workstation.

In terms of interior design features, the offices have glass walls, carpeted floors, and two wooden staircases. The rest of the space has painted cobblestone walls, two large windows, and an artificial garden with stone flooring on one side. Within the selected area were 12 workstations, for which 18 measurement points were located for the acoustic study (Figure 1).

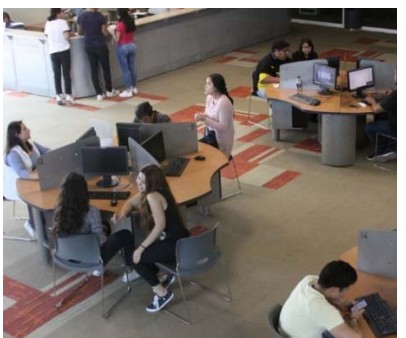 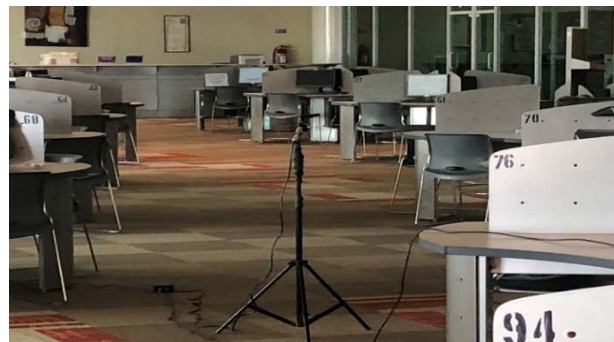

**Figure 1.** Layout of the computer service area.

The acoustic analysis consisted of an environmental noise study and an intelligibility acoustic study. The environmental noise study was carried out according to Mexican standards using a type 2 sound level meter. In contrast, the acoustic study was performed according to ISO international standards using a microphone and two omnidirectional loudspeakers. In addition, two sound recordings were used.

The environmental noise analysis was performed to identify the sound conditions of the office, and the acoustic analysis was carried out to evaluate the intelligibility in the different places of the selected area. In Mexico, there is no specific regulation on acoustics in open-plan office environments since Mexican regulations refer to the evaluation and control of noise in workplaces. The environmental noise study was carried out based on NOM-011-STPS-2001 Noise [26].

The soundscape of the office studied has stable noise. Based on NOM-011 Noise, the environmental noise study was conducted with the Sound Pressure Gradient method, for which two trajectories were established (linear and transverse), and sound pressure levels were measured with a type 2 sound level meter at different measurement points, looking for differences of 3 dB to identify changes in sound power in the area.

The measurement points were assigned according to NOM-011, which requires that dimensional measurements be taken at the beginning of the study and then the space be divided into a grid. In this study, the grid was formed with 46 measurement points with 3 m. On the other hand, the acoustic analysis was performed with architectural acoustic design techniques by estimating the reverberation of the enclosure. For the analysis of intelligibility, the specialized software Dirac® version 6 was used. To determine the architectural acoustic design criteria, the reverberation characteristics of the enclosure were estimated. First, the

size of the enclosure was calculated: $P$ (m) is the perimeter, $A$ (m$^2$) is the total surface, and $V$(m$^3$) is the volume.

Then, according to the construction materials and furniture, the absorption coefficients were determined using Sabine's formula to determine the reverberation time, $RT$ (1), for which the average absorption coefficient, $\bar{\alpha}$ (2), was calculated. With the estimated $RT$, an acoustic survey of the area was performed.

$$RT = \frac{0.161\ V}{\bar{\alpha} S_t} \tag{1}$$

$$\bar{\alpha} = \frac{A_{tot}}{S_t} \tag{2}$$

Sound absorption is an element used for architectural acoustic design. The degree of sound absorption of materials is estimated by the absorption coefficient $\alpha$, which is defined as the ratio between the energy absorbed by the material and the energy incident on it. It is important to consider that the $\alpha$ values are directly related to the physical properties of the material and vary with each frequency. Subsequently, to compare methods for measuring intelligibility, the Dirac$^\circledR$ acoustic software is used to measure room acoustic parameters in field or laboratory applications according to ISO 3382 (room acoustics), ISO 18233 (analysis methods), and IEC 60268-16 (speech intelligibility) [27].

The acoustic analysis was performed with objective and subjective methods. The objective methods were analyzed for the effects of sound factors such as the type of sound, the location of the sound sources, and the measurement points by means of acoustic reverberation and speech transmission parameters. With the subjective methods, the speech quality assessments were analyzed in terms of psychoacoustic parameters at each of the measurement points within the workstation area.

The comparison of intelligibility measurement methods was performed considering four variables at each measured point. Two variables for the objective method were the $RT$ and $STI$, and two variables for the subjective method were the $\%ALCons$ and the subjective speech quality rating. All variables were obtained from the Dirac$^\circledR$ software. The subjective speech quality analysis method is based on ISO 3382-3, which ranks the $STI$ and $\%ALCons$ values on a categorical rating scale, Table 1. On the other hand, two factors of interest were used to compare the methods: the location of the measurement points, the sound sources, and the sound conditions under which the study was carried out.

**Table 1.** Relationship between *STI* and *%ALCons* and the subjective assessment of the degree of intelligibility.

| Speech Intelligibility | Speech Transmission Index *RASTI* | Percentage Articulation Loss of Consonants (*%ALCons*) |
|---|---|---|
| Bad | 0–0.30 | 27–46.5 |
| Poor | 0.30–0.45 | 12–24.2 |
| Fair | 0.45–0.60 | 5.3–11.4 |
| Good | 0.60–0.75 | 1.6–4.8 |
| Excellent | 0.75–1 | 0–1.4 |

The assignment of the measurement points was obtained from a sample of 18 of the 46 measurement points on the grid used in the environmental noise analysis. The location of the sound sources was determined using guidelines from the ISO 3382-3 standard. This involved utilizing data obtained from the reverberation of the room and calculating the Minimum Distance (3) *Dmin* [18].

$$Dmin = 2\sqrt{\frac{V}{cRT}} \tag{3}$$

The following factors were used for this study: (a) location of the sound sources—Source A was located outside the work area to meet the *Dmin* requirements with respect to the workstations, while Source B was located inside the study area with distances less than the *Dmin* with respect to the workstations; (b) location of the measurement points within the work area; and (c) types of sounds emitted—silence, which represents the office soundscape, and a sound recording of office noise, which is a recording with real office sounds.

Additionally, the treatment of the factors was a random combination of the sounds emitted from each sound source. This formed four sound conditions: emission of office noise by source A, emission of office noise by source B, emission of office noise by both sources AB, and not emitting the background noise 0, i.e., leaving the soundscape of the office. Additionally, to control the experiment, only one frequency band was used, establishing it as a block variable because this frequency is the one that provides the greatest contribution to the intelligibility of the word [1].

Finally, the data obtained from the acoustic analysis, with the experimental design, were analyzed using a General Linear Model (GLM) to determine the behavior of the statistically significant acoustic parameters. Because some parameters did not fit a normally distributed behavior, these were adjusted with the Box–Cox power transformation for normality in the residuals.

At the same time, to represent the variation in sound conditions, the sound factor was nested to represent the variation of the sound combinations emitted during the study. On the other hand, with the GLM, the correlation coefficients (β) were determined to subsequently determine the effect size using Cohen's test for the mean values and correlation coefficients (d), as well as for the regression (f) of the factors with significant changes during the study [28].

## 3. Results

The results of the study were obtained from two analyses of the acoustics of space. The first study was of the environmental noise for acoustic recognition in the enclosure, showing the behavior of the sound pressure level (*SPL*) at various points of the office. Subsequently, an acoustic study was carried out to determine the intelligibility (*STI* and *%ALCons*). In addition, with this study, the categorization of the speech quality at each point measured in the office was performed. With the data obtained, a statistical analysis was performed to recognize the behavior of each parameter and make a comparison between them.

### 3.1. Environmental Noise Study

The environmental noise study was conducted under Mexican regulations using the Sound Pressure Gradient method [26]. The Sound Pressure Gradient method evaluates the sound environment. The studied area of the library showed stable ambient noise with an *SPL* range between 51 and 55 dB. The sound behavior is represented with a heat map, which shows that the areas near the cubicles with glass walls increase *SPL* while the areas near the stairs decrease *SPL* (Figure 2).

In this study, nine points were measured in five zones with changes greater than 3 dB. Within the work area, four points were detected with changes in sound pressure, while the remaining points were in an unoccupied area. The result obtained from the noise recognition study was consistent with ambient noise conditions in open-plan offices, where the background noise is below the established sound level limits (80–90 dB). However, this does not imply that workers do not feel affected in terms of their well-being or their perception of unproductivity [29].

The results of the environmental noise study identified the following characteristics of the open-plan office: (a) The area with the lowest *SPL* = 51 dB had a radius of 6 m located near the wooden stairs; (b) the area with the highest *SPL* = 55 dB had a radius of 3 m recorded by the corner near the cubicles with glass walls; (c) most of the measured points presented a range of 52–54 dB; and (d) the areas where the *SPL* changes (3 dB) occurred were located at the center of the work area and the edges near the unoccupied space of the office.

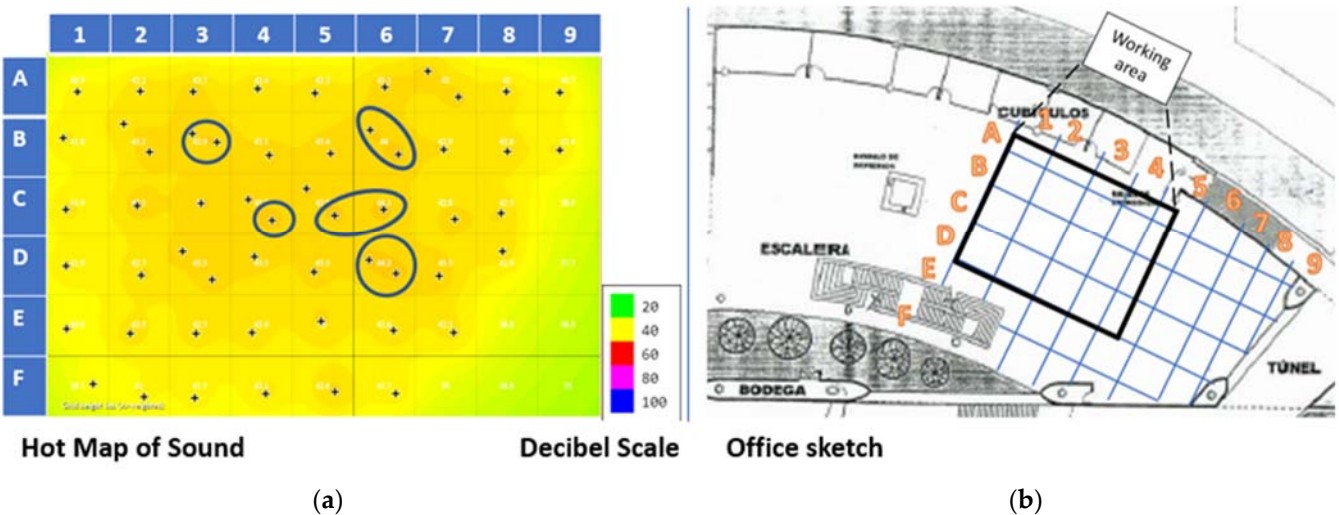

Hot Map of Sound     Decibel Scale     Office sketch

(**a**)          (**b**)

**Figure 2.** Sound heat map and office arrangement in the computer service area: (**a**) Sound heat map, indicating the points where changes in sound pressure levels occurred; (**b**) Sketch of the office, schematization on the plan of the area selected for the study.

Subsequently, the architectural acoustic analysis was performed to estimate the Reverberation Time (*RT*) with Sabine's Formula (4) and the acoustic absorption coefficients of the materials of the office and the enclosure. It is important to emphasize that the acoustic absorption coefficients vary in each frequency band. Still, for this case, we only worked with the 2 kHz frequency because this frequency contributes 34% to speech intelligibility and is the frequency with the highest contribution to speech understanding.

Consequently, Sabine's formula resulted in a *RT* = 0.8 s; furthermore, at mid and high frequencies, a diffraction effect ($\bar{\alpha} > 1$) was detected, particularly for the 2 kHz frequency, where the estimated total sound absorption was $\bar{\alpha}$ = 2.41 [1]. The diffraction effect is explained by the interior design of this office. Since the design has cubicles of glass that are semi-closed, this openness causes echoes to be generated in different areas of the open-plan office. Due to this diffraction effect, it was not possible to determine the critical distance to locate the sound sources, which are commonly used. Therefore, it was replaced with the minimum distance, which for this area was *Dmin* = 9.29 m for frequency 2 kHz [18].

*3.2. Acoustic Analysis of Intelligibility*

The intelligibility analysis was performed with Dirac® software, which generates data according to ISO 3382, a Reverberation analysis, and the classification according to speech quality categories. The data collection for measuring reverberation and intelligibility was performed using an experimental design for the combination of the factors of interest: the location of the measurement points and sound sources, the type of sound, and combining the emission of office noise.

For the experimental design, 18 measurement points were considered within the work area; with respect to the sound source, five positions were considered, depending on *Dmin*, i.e., three locations outside the work area were considered to comply with *Dmin* (Source A: P1A, P1B, and P1C) and two other locations within the work area that do not comply with *Dmin* (Source B: P2A and P2B).

While the type of sound was considered, the soundscape was nominated as silence, and the recording of an office was nominated as office noise. For the experimental design, for the sound conditions, the combinations of the sound sources for the emission of the sound types were randomized (0: Silence, A: External source, B: Internal source, and AB: Both sources). Additionally, this combination was used to measure the intelligibility at each measurement point (Figure 3).

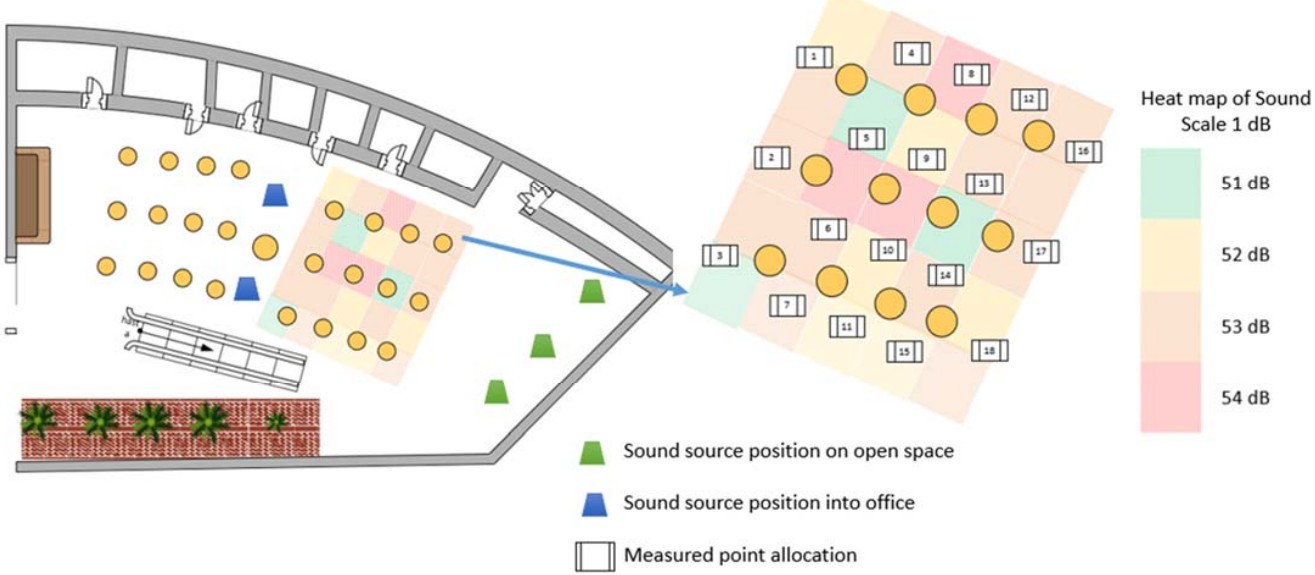

**Figure 3.** Scheme of the sound factors of the study. This schematic represents the sound heat map, with a color scale for the position of each measurement point within the working area.

### 3.2.1. Analysis by Objective Methods

For the acoustic analysis with objective methods, the Dirac® software was used with reverberation and speech parameters according to ISO 3382, from which some were selected for analysis. The reverberation parameters used were early and late sound energy ratio with sound decay measurements below −60 dB because sound behavior in offices is generally explained by early energy. The reverberation parameters used in this study were the Early Decay Time (*EDT*), which is based on the first and earliest reflections for a sound decay of −10 dB.

Other parameters used were reverberation time (*RT*), for a sound decay of −60 dB and T30, for a sound decay of −30 dB, the latter being recommended for offices. According to ISO 3382, the speech parameters used for analysis were Speech Transmission Index (*STI*) and Loss of Articulation of Consonants *(%ALCons)*. The *%ALCons* was a parameter used to represent the subjective methods apart from the subjective rating categories, which are also provided by the Dirac® software.

After the experimental design, 408 data were obtained for each parameter (*EDT*, *RT*, *T30, STI,* and *%ALCons*) with all combinations of the factors of interest. A GLM analysis was performed to explain the behavior of the data. Reverberation parameters did not present significant changes with the factors of interest of the study. Only three measurement points presented changes in the average measured time:

- Point 8, when office noise was emitted from sources located outside the work area with a medium effect, the *EDT* averaging time decreased ($\beta = -1E^{-6}$, d = 0.63). This point is in an area with *SPL* changes and near the glass walls, which causes sound reflection.
- Point 15, the *EDT* varied when the office noise was emitted, but depending on the location of the external sound source was the contribution to the average time. This means that when the emission came from the position to the right near the wall (L1A), there was a small effect in the decrease in the *EDT* ($\beta = -1E^{-6}$, d = 0.20). On the other hand, if the emission came from the position to the center of the window (L1B) with a large effect, the *EDT* times increased ($\beta = 2E^{-5}$, d = 1). This point is in a zone without *SPL* changes and close to an open space.
- Point 16, the *EDT* presented changes in the meantime when no sound was emitted from the external sound source positioned at the center of the window (L1B). This decreased the *EDT* ($\beta = -1E^{-6}$, d = 1). This point was located close to glass walls but abutted the unfurnished open space.

On the other hand, intelligibility parameters presented significant changes with respect to some factors. The speech transmission analysis showed that STI changed its value with a small effect when assigning measurement point ($p = 0.009$, f = 0.33) and with the location of the sound source ($p < 0.001$, f = 0.28), while STI changed with a large effect when varying the type of sound ($p < 0.001$, f = 0.55) (Figure 4).

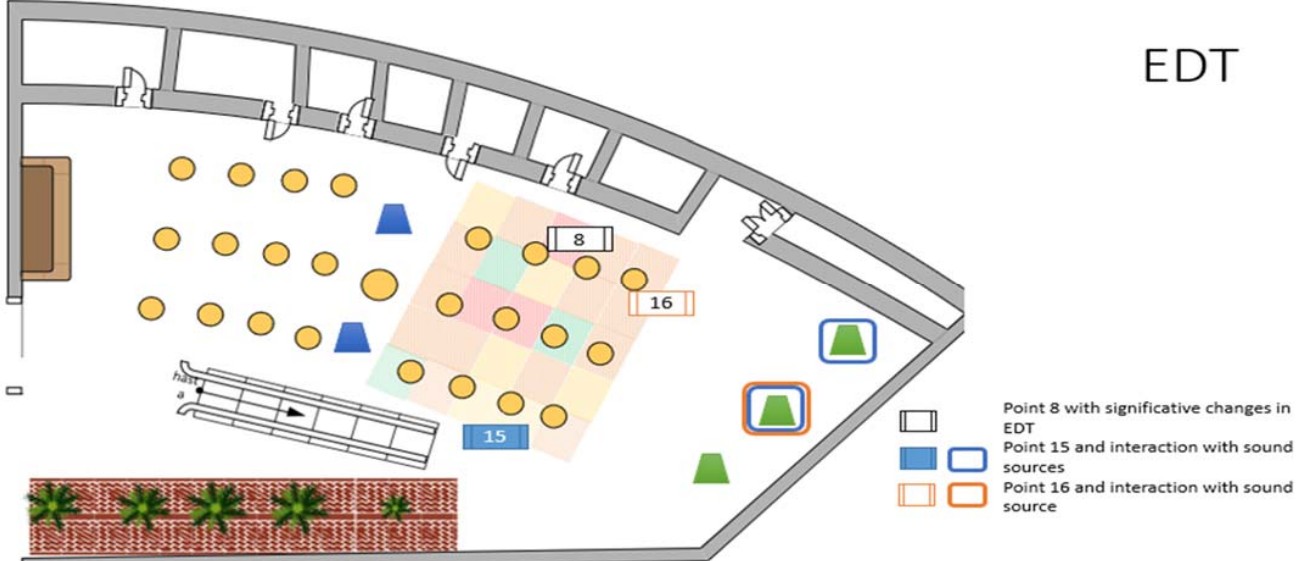

**Figure 4.** Points with significant changes in *EDT*. This diagram shows the points and sound sources with significant changes in the average values during the study.

During the study, the mean value of the intelligibility levels was *STI* = 0.68, indicating a good overall speech quality level for the room. In addition, the *STI* presented specific significant changes (Figure 5). Office noise emission was favored with a negligible effect, with an *STI* = 0.71 ± 0.14 ($p < 0.001$, d = 0.19); Point 5 varied the levels averaging with a medium effect and an *STI* = 0.77 ± 0.16 ($p = 0.004$, d = 0.56); Point 5 varied with a medium effect and an *STI* = 0.77 ± 0.16 ($p = 0.004$, d = 0.56); Point 12 varied with a negligible effect and an *STI* = 0.71 ± 0.14 ($p < 0.001$, d = 0.19); Point 12 varied with a medium effect, averaging an *STI* = 0.59 ± 0.16 ($p = 0.004$, d = 0.56). This point was located next to the glass walls and in an area without changes in *SPL*; Point 13, with a small effect presented at *STI* = 0.59 ± 0.18 ($p = 0.005$, d = 0.19), was located at the center of other cubicles and in an area without changes in (Figure 5).

In addition, the characteristics of the materials influenced the correlation between the assignment of measurement points and *STI* levels. Here are some examples:

- In areas with SPL changes, point 5, surrounded by worktables, had a medium effect, and increased *STI* levels (β = 0.0692, d = 0.56). However, point 8, which is close to sound-reflecting glass walls with a negligible effect, caused a decrease in *STI* levels (β = −0.0673, d = 0.06) when interacting with office noise emission.
- In areas with stable *SPL*, some measurement points stand out:
    - Point 12, close to glass walls and a corridor, showed a medium effect, a decrease in *STI* levels (β = −0.0924, d = 0.56), which continued this trend up to Point 13, decreasing *STI* levels (β = −0.0894, d = 0.56).
    - Point 16 stands out because interacting with the office noise emission produced a negligible decrease in *STI* levels (β = −0.0673, d = 0.06). However, at Point 3, when the office noise emission came from external sources, it presented a medium effect in decreasing *STI* levels (β = −0.1279, d = 0.57).
- The type of sound was another main effect that influenced the behavior of the *STI* level, particularly office noise:

○ When the emission came from sources outside the work area, there was a negligible decrease in *STI* levels (β = −0.0499, d = 0).

○ When the emission came from both sound sources with a negligible effect, there was a decrease in *STI* levels (β = −0.048, d = 0).

- The location of the external sound source, Source A, also influenced *STI*-level behavior:

  ○ When the external sound source located to the right (L1A) was combined with any internal sound source to emit both office noise, there was a medium effect in decreasing *STI* levels (β = −0.0405, d = 0.36).

  ○ When the office noise emission came from either source location A, Point 8 near glass walls had a small effect on decreasing *STI* levels (β = −0.1533, d = 0.24). In contrast, Point 10 located between work modules near wooden stairs, had a large effect on decreasing *STI* levels (β = −0.109, d = 0.60) (Table 2).

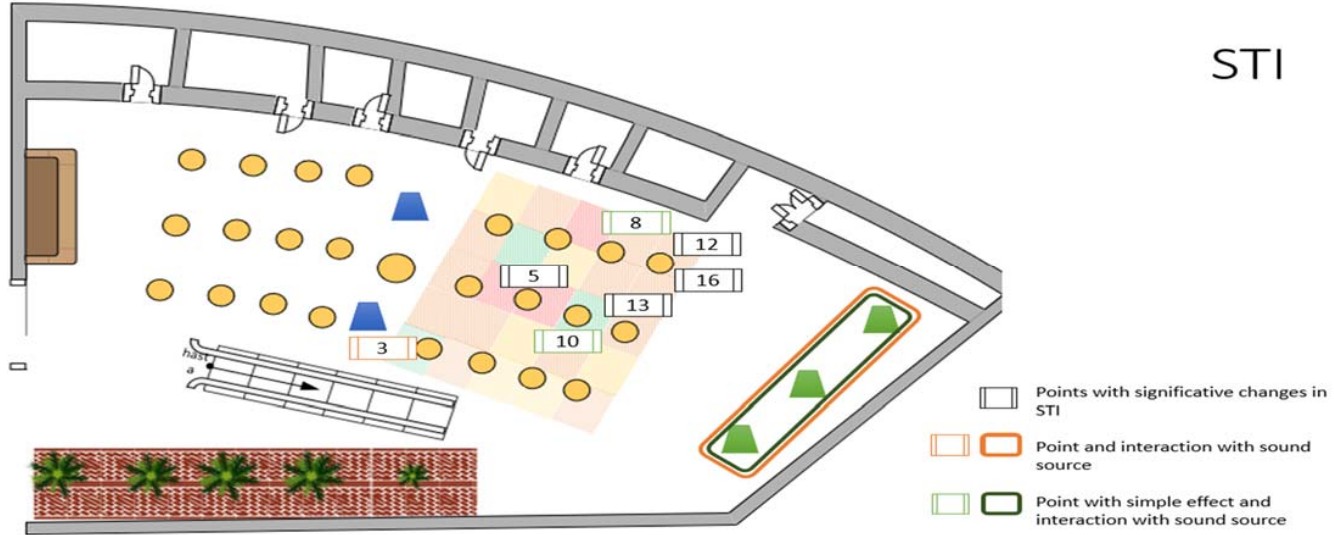

**Figure 5.** Points with significative changes in *STI*. This diagram shows the points and sound sources with significant changes in the average values during the study.

**Table 2.** Relationship between *STI* and study factors, α = 0.05 (ON = Office Noise record, S = Simple effect, I = Interaction).

| | Factors | | | | *STI* | | | |
|---|---|---|---|---|---|---|---|---|
| Point | *SPL* Zone | Type of Sound | Condition | Type of Effect | β Coefficient | *p*-Value | Mean | Stand Desv |
| | | ON | | S | 0.079 | <0.001 | 0.71 | 0.144 |
| | | ON | Out area | I | −0.050 | <0.001 | 0.67 | 0.133 |
| | | ON | Both area | I | 0.048 | <0.001 | 0.76 | 0.147 |
| 3 | Stable | ON | Out area | I | −0.128 | 0.044 | 0.61 | 0.085 |
| 5 | Changes | | | S | 0.069 | 0.029 | 0.77 | 0.157 |
| 8 | Changes | ON | | I | −0.067 | 0.034 | 0.66 | 0.152 |
| | Changes | ON | Out area | I | 0.153 | 0.001 | 0.76 | 0.161 |
| 10 | Changes | ON | Out area | I | −0.109 | 0.015 | 0.53 | 0.074 |
| 12 | Stable | | | S | −0.092 | 0.004 | 0.59 | 0.16 |
| 13 | Stable | | | S | −0.089 | 0.005 | 0.59 | 0.181 |
| 16 | Stable | ON | | I | −0.064 | 0.044 | 0.7 | 0.167 |

### 3.2.2. Subjective Analysis

The subjective methods were based on the subjective assessment of intelligibility based on the Statistical Acoustic Theory. For the subjective analysis, the consonant loss parameter was used to evaluate intelligibility by means of a quality scale. The scale uses the Puetz formula for *%ALCons* modified by Farrel Becker, expressing intelligibility between 0–1.

These percentages generate a classification for subjective evaluation, with categories of excellent, good, fair, and poor [14].

The subjective methods were also analyzed using the Dirac® software, with which the estimated *%ALCons* results were obtained. This parameter had a special treatment since the *%ALCons* values obtained had to be adjusted to a Lognormal distribution to perform the GLM analysis. For this adjustment, a Box–Cox transformation ($\lambda = 0$) was used [28]. The subjective rating was analyzed using the Dirac® program, which yielded the *%ALCons* values to be subsequently classified according to the subjective categorization of speech quality (Table 1).

The *%ALCons* parameter presented the influence of some factors in the variation of its average percentages. The factors of interest that influenced the *%ALCons* were the assignment, the location of the measurement points ($p = 0.006$, f = 0.33), and the type of sound ($p < 0.001$, f = 0.57). In addition, the interaction between some factors presented some changes in this parameter. For example, a medium effect was found for the influence of sound nesting on sound condition ($p < 0.001$, f = 0.30) and the interaction of this with measurement point assignment ($p = 0.04$, f = 0.40). Sound nesting explains that the variation in sound emission has an influence depending on the type of sound transmitted and how it interacts at each point where the acoustic parameters are measured.

In this study, the general average was *%ALCons* = 6.51, which can be classified as adequate quality for the intelligibility of this enclosure. Some specific points presented some changes, such as the case of Point 5 with a large effect that had a *%ALCons* = 3.71 ± 1.59 ($p = 0.024$, d = 1); Point 10 had a large effect with *%ALCons* = 7.96 ± 2.82 ($p = 0.039$, d = 0.73); Item 12 had a large effect with *%ALCons* = 9.78 ± 2.12 ($p = 0.003$, d = 1); and Item 13 had a large effect with *%ALCons* = 10.21 ± 2.47 ($p = 0.007$, d = 1) (Figure 6).

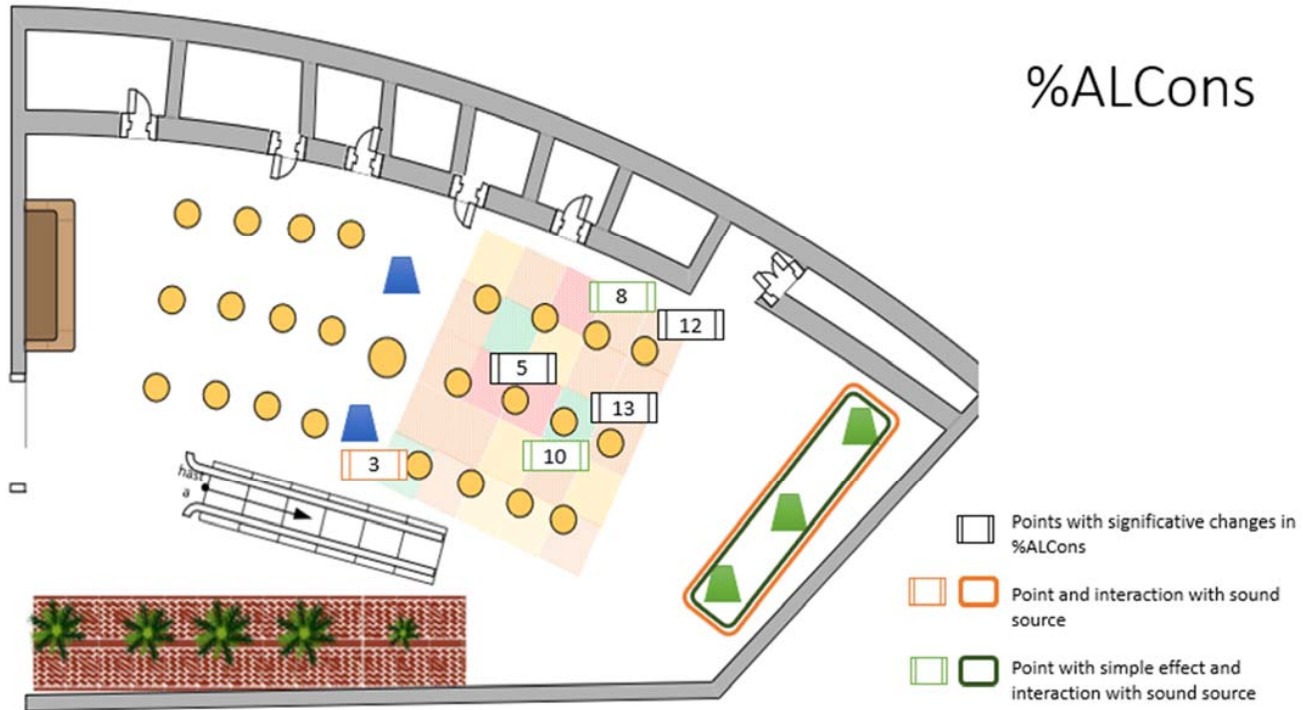

**Figure 6.** Points with significative changes in *%ALCons*. This diagram shows the points and sound sources with significant changes in the average values during the study.

In addition, some factors stood out in the effect on the loss of consonants, particularly with the sound conditions. For example, the type of sound with office noise presented a large effect on the average *%ALCons* = 1.84 ± 0.52 ($p < 0.001$, d = 1). On the other hand, the origin of the emission also influenced the loss of consonants. In this case, when the office

noise emission came from an external sound source (A) with a small effect, it increased the %*ALCons* (β = 0.281, d = 0.41). In contrast, when the sound came from both sound sources (AB) with a large effect, it decreased the percentages (β = 0.282, d = 1) (Table 3).

**Table 3.** Relationship between %*ALCons* and study factors, α = 0.05 (ON = Office Noise record, S = Simple effect, I = Interaction).

| | | Factors | | | %*ALCons* | | | |
|---|---|---|---|---|---|---|---|---|
| Point | *SPL* Zone | Type of Sound | Condition | Type of Effect | β Coefficient | *p*-Value | Mean | Stand Desv |
| | | ON | | S | −0.436 | <0.001 | 1.84% | 0.52% |
| | | ON | Out area | I | 0.281 | <0.001 | 5.69% | 0.10% |
| | | ON | Both area | I | −0.282 | <0.001 | 3.24% | 0.74% |
| 3 | Stables | ON | Out area | I | 0.738 | 0.029 | 6.91% | 0.95% |
| 5 | Changes | | | S | −0.381 | 0.024 | 3.71% | 1.59% |
| 8 | Changes | ON | | I | 0.370 | 0.028 | 6.82% | 1.88% |
| | Changes | ON | Out area | I | −0.831 | 0.001 | 4.09% | 0.85% |
| 10 | Changes | | | S | 0.348 | 0.039 | 7.96% | 2.82% |
| | Changes | ON | Out area | I | 0.556 | 0.02 | 10.59% | 0.98% |
| 12 | Stables | | | S | 0.510 | 0.003 | 9.78% | 2.12% |
| 13 | Stables | | | S | 0.455 | 0.007 | 10.21% | 2.47% |

In addition, some measurement points, when interacting with certain conditions, presented some effects: Point 3, when the emission from an external source (A) had a large effect, increased the percentages (β = 0.738, d = 1). Point 8, when sound noise was emitted, had a large effect, increasing the percentages (β = 0.37, d = 1). However, if the emission came from an external source (A) with a large effect, it decreased %*ALCons* (β = −0.831, d = 1).

Finally, for the subjective methods, the classification of speech quality was used. The overall averages were classified as the average values for *STI* with a good quality level (*STI* = 0.68) and for %*ALCons* with an adequate quality level *(%ALCons* = 6.51). This indicates lower quality recorded for %*ALCons*. This behavior is also present in the measurement points with significant changes.

Initially, the *STI* presented three categories in the 11-point classification. The category with the highest quality was excellent with 27% of the points measured, followed by good with 45% and adequate with 27%. However, for %*ALCons*, there were only two categories for the 11 points with significant changes. With this parameter, the quality decreased, classifying only the good category with 36% of the points measured and the adequate category with 64%. The results of the speech quality indicated that the *STI* was classified with higher quality, mostly presenting a good category. However, for %*ALCons*, the classification was more neutral, presenting mostly an adequate level.

### 3.3. Psychoacoustic Parameter Comparison

In this study, effect size was the technique for comparing the methods for measuring intelligibility. For this reason, the effect size was evaluated with Cohen's test. In this case, the d-test for mean values and correlation coefficients (adjustable to Student's *t*-test) and the f-test for factor analysis (adjustable to regression values) were used. The effect size was estimated with the points measured with statistically significant results (α < 0.05).

According to the results obtained, it should be noted that for some points, the size of the effect coincides with the three parameters analyzed (Table 4). Point 10 has a medium effect on the three parameters. In the case of *EDT* and %*ALCons*, it increased its values, and in the case of *STI*, it decreased its levels, which indicates that at that point, the quality of intelligibility could worsen.

**Table 4.** Effect size for the relationship between *EDT*, *STI*, *%ALCons*, and study factors, $\alpha$ = 0.05 (S = Stable, C = Changes, ON = Office Noise record, OA = Out area, BA = Both areas, I = Insignificant, Sm = Small, M = Medium, L = Large).

| Factors | | | | | EDT Effects | | STI Effects | | %ALCons Effects | |
| Point | SPL Zone | Type of Sound | Condition | Sound Source | Effect Size (d) | β Coefficient | Effect Size (d) | β Coefficient | Effect Size (d) | β Coefficient |
|---|---|---|---|---|---|---|---|---|---|---|
| | | ON | | | | | 0.19　I | 0.079 | 1　L | −0.436 |
| | | ON | OA | | | | 0　I | −0.050 | 0.41　M | 0.281 |
| | | ON | BA | | | | 0　I | 0.048 | 1　L | −0.282 |
| 3 | S | ON | OA | | | | 0.57　M | −0.128 | 1　L | 0.738 |
| 5 | C | | | | | | −0.56　M | 0.069 | 1　L | −0.381 |
| 8 | C | ON | | | | | −0.06　M | −0.067 | 0.82　L | 0.370 |
| | C | ON | OA | | | | 0.24　Sm | 0.153 | 1　L | −0.831 |
| 10 | C | | | | 0.63　M | $-1\text{E}^{-6}$ | 0.6　M | −0.109 | 0.73　M | 0.348 |
| | C | ON | OA | | | | | | 0.93　L | 0.556 |
| 12 | S | | | | | | 0.56　M | −0.092 | 1　L | 0.510 |
| 13 | S | | | | | | 0.56　M | −0.089 | 1　L | 0.455 |
| 15 | S | ON | OA | L1A | 0.2　S | $-1\text{E}^{-6}$ | | | | |
| | S | ON | OA | L1B | 1　L | $2\text{E}^{-5}$ | | | | |
| 16 | S | ON | | | | | −0.06　I | −0.064 | | |
| 16 | S | ON | OA | L1B | 1　L | $-1\text{E}^{-6}$ | | | | |

On the other hand, for the EDT, Point 10 was the only point with significant changes in its average values that coincided with the other variables studied. Nevertheless, *STI* and *%ALCons* parameters coincided in affecting the same measured points. However, the effect size was larger in the points measured with *%ALCons* than *STI*, indicating that the factors with which the study was conducted describe *%ALCons* with greater magnitude.

Particularly, the factors related to the locations presented the following behaviors: the measurement points showed a medium effect for both parameters *STI* and *%ALCons*. As for the sound source, there was a moderate effect observed when modifying the average levels of *STI*. However, in the case of *%ALCons*, the effects were presented when interacting the sound source with the type of sound. For the statistical analysis, the randomness of the sounds emitted was represented by nested variables, i.e., for the LGM the sound type variable was nested with the sound condition (i.e., the origin of the sound), this combination of variables was significant for %ALCons with a medium effect; and when interacting this nested variable, the effect increased to large. In other words, for %ALCons there is an influence in the change of its average percentages when varying the origin of the sound emission with the place where the sound measurement is registered (Table 5).

**Table 5.** Effect size for the relationship between *STI*, *%ALCons*, and study factors, $\alpha$ = 0.05.

| Factors | Effect Size (f) | | STI p-Value | Effect Size (f) | | %ALCons p-Value |
|---|---|---|---|---|---|---|
| Measurement point location | 0.33 | Medium | 0.009 | 0.33 | Medium | 0.006 |
| Sound source location | 0.33 | Medium | <0.001 | | | |
| Type of sound | 0.55 | Large | <0.001 | 0.57 | Large | <0.001 |
| Sound source location and type of sound (nested) | | | | 0.30 | Medium | <0.001 |
| Sound source location, type of sound (nested), and measurement point location | | | | 0.40 | Large | 0.04 |

On the other hand, with respect to the classification of speech with intelligibility parameters, the effect sizes were different. Specifically, of the 11 measurement points with significant changes in ITS, three categories of speech quality were presented, 45% in the Good category, most of these points indicated a negligible effect; the remaining 27% were equally classified as Excellent with a negligible, small and medium effect, while the other 27% as Adequate, in all cases with a medium effect. In contrast, for %ALCons of the 11 measurement items with significant changes, 64% were classified as Adequate, mostly with a large effect and some items with a medium effect, the remaining items were classified as Good, all with a large effect.

## 4. Discussion

In this case study, several elements that make up the soundscape of an open-plan office were examined. The evaluation focused on the main and simple effects, the interactions related to the different types of sound, the location of both sound sources, and the points where acoustic measurements were taken.

Braat-Eggen et al. [24] indicated that realistic scenarios with different acoustic parameters should be explored. Yadav et al. [8] specified that the acoustic analysis of the soundscape should be approached with psychoacoustic parameters. Therefore, psychoacoustic parameters of intelligibility were analyzed in this case study. Specifically, the acoustic characteristics of reverberation and speech transmission were studied. The reverberation analysis indicated that the interior design influences the acoustics of the space. In this case, the construction materials and furniture generate a diffraction effect [30].

The diffraction effect is related to interior design features and their influence on space acoustics. We detected it at measurement points with significant changes in *EDT* near areas with interior design features. For example, *EDT* presented medium effect changes at measurement points near glass walls during this study. On the contrary, measurement points near open areas where the study concluded showed a large effect with respect to the correlation coefficients β, this indicates that the acoustic qualities of the enclosure influenced the analyzed behaviors.

This type of behavior can be considered in architectural acoustic design with virtual tools to anticipate such design effects. Nowoswiaf and Olechowska [31] followed experimental techniques to simulate *RT*-based room acoustics. Furthermore, according to data presented by Trocka and Jablonska [32], guidelines and recommendations should be made for the architectural acoustic design of open-plan offices.

For countries that do not have an intelligibility standard, these studies can support the effects of architectural acoustic design on performance. This was demonstrated by Park and Haan [33], who focused on school classrooms only using *RT* as an indicator to assess reverberation. Their study provides an acoustic performance standard for classrooms. With this study, we can extend to study and consultation areas in educational centers.

Architectural acoustic design influences the improvement of the acoustic conditions of spaces for the perception of comfort and well-being of users. For example, the critical distance, which is based on the ratio between RT and SPL to determine the distracting radius, this distance is the one that influences the perception of lack of privacy of open office users. Hongisto and Keränen [34] propose a classification scheme for critical distance estimation based on speech attenuation performance. This facilitates the interpretation of acoustic measurements to determine comfort distance based on ISO 3382-3 [35].

This may contribute to what Braat-Eggen et al. [24] suggest about the influence of changes in the reverberation times of a room on the impact of intelligibility on writing activities. Regarding speech transmission, *STI* is one of the main parameters used to evaluate the acoustic characteristics of the room. In the case of open-plan offices, *STI* is used to evaluate the intelligibility levels that allow hearing speech clearly. However, it is known that this clarity of speech in background conversations can cause discomfort to users in open-plan offices [36].

In addition, following the architectural design requirements, another psychoacoustic parameter for speech transmission analysis is *%ALCons*. It is part of the subjective analysis to assess how people perceive the quality of intelligibility at different points in the room space [16]. During this study, the results indicated that sound type influences psychoacoustic parameters and measurement point assignment, although the effects were small.

As expected, the results showed an inverse relationship between *STI* and *%ALCons*, but the effect size was different. This change is attributable to the behavior of the parameter values obtained. Based on the statistical analyses, it could be determined that the *STI* parameter showed normal behavior, while the *%ALCons* parameter showed Lognormal behavior. The Lognormal behavior of *%ALCons* represents approximately the scale for describing the extent to which listeners perceive speech. [37]. According to studies, speech

tends to have Lognormal behavior due to the structure of phonemes and words. As explained by Torre et al. in their statistical learning studies focused on acoustic elements with long-tailed distributions, this study gives a line to contribute to the existing body of research on the application of the Lognormal Law proposed for the acoustic linguistic units [38].

In addition, the presented study provides information for the interpretation of the behavior of psychoacoustic parameters in a real working environment. The selected diaphanous office offers a perspective for using work and study areas. Despite being situated in a library, this diaphanous office was a consultation center where users could interact without total silence. This is consistent with the findings of Molesworth et al. [39]. They observed that, even in a laboratory setting, it was possible to work in a more realistic environment where the type of sound influences the performance of activities, especially writing and consultation.

The actual environment in which we worked during this study coincides with that proposed by Zoghbi et al. [40]. This could be related to interpersonal perceptions of the psychosocial conditions of this type of work area. Therefore, we could also continue analyzing background noise control and its effects on stress by monitoring cognitive activities in spaces such as diaphanous offices [40–42]. Finally, it should be noted that according to what Altomante et al. [43] propose, a space design agenda should be generated to promote well-being based on IEQ. In this case, we focus especially on acoustic comfort. Glean et al. [44] proposed that acoustic solutions in an open-plan office should come from a people-centered acoustic environment [9,45].

## 5. Conclusions

The study of intelligibility was chosen to present an overview of the perception of background noise by users of open-plan offices to explain the use of psychoacoustic parameters in the open-plan office. This paper presents a case study in which objective and subjective methods are used to assess intelligibility to identify the effects of sound conditions with respect to factors such as the location of sound sources and with different types of sounds in an open plan office.

During the case study different results were obtained; initially, an Environmental Noise study was performed, which is a required study in Mexico to control the noise level in any type of work centers. Subsequently, an acoustic analysis of intelligibility was performed for an open plan office by objective and subjective methods using Dirac software. The open plan office analyzed in this study is representative of the interior design style in Mexico, but its architectural design is not, so it would be advisable to carry out more studies on this type of spaces, since the acoustic design is usually carried out during the design stages and, in most cases, is not followed up when the office is implemented. In addition, it is important to consider in acoustic design the effects of semantics and subjective speech evaluations on performance, which not only produce a perception of discomfort for users of open-plan offices.

This study contributes three approaches to improve comfort in open offices with positive effects on the perception of well-being and productivity of workers. On the one hand, architects, designers, and contractors should focus on the architectural acoustic design to create spaces where intelligibility generates comfortable and healthy spaces, through optimal conditions to mitigate the noise that is a distractor in its users. Meanwhile, for workplace safety, guidelines and recommendations for noise level management are outlined to optimize the effect of acoustic conditions on the cognitive performance of workers.

In addition, for companies to apply personal strategies to increase productivity, studies can focus on linguistics to outline studies of expected speech behavior and its effect on written activities. This research explores acoustic design in open offices, for which in this field it is suggested to follow up acoustic design interventions with longitudinal studies to evaluate long-term effects in specific workplaces, as well as to deepen the impact of different acoustic design elements. It is also suggested to evaluate the influence of acoustic

design in different types of office activities and work tasks, in different specific workstations or in meeting or collaboration areas.

**Author Contributions:** Conceptualization, M.P.S.-R. and G.I.-M.; methodology, M.P.S.-R. and G.I.-M.; software, J.A.Y.-A.; validation, M.P.S.-R., J.A.Y.-A. and G.I.-M.; formal analysis, M.P.S.-R. and J.A.Y.-A.; investigation, M.P.S.-R.; resources, G.I.-M. and E.A.M.-G.; data curation, M.P.S.-R.; writing—original draft preparation, M.P.S.-R.; writing—review and editing, M.P.S.-R., J.A.Y.-A and G.I.-M. visualization, M.P.S.-R. and G.I.-M.; supervision, G.I.-M.; project administration, M.P.S.-R. and G.I.-M.; funding acquisition, G.I.-M. All authors have read and agreed to the published version of the manuscript.

**Funding:** This research has not received external funding.

**Institutional Review Board Statement:** Not applicable.

**Informed Consent Statement:** Not applicable.

**Data Availability Statement:** Not applicable.

**Acknowledgments:** The authors would like to thank the manager and staff of the Infoteca of the Universidad Autonoma de Coahuila, who offered their collaboration to carry out the measurements in the facilities.

**Conflicts of Interest:** The authors declare no conflict of interest.

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
