# Peer review of "Open-Plan Offices: Comparison of Methods for Measuring Psychoacoustic Intelligibility Parameters"

_applsci, doi:10.3390/app13158650_

Round 1

Reviewer 1 Report

The manuscript titled “Open-plan offices: comparison of methods for measuring Psychoacoustic intelligibility parameters” was reviewed. The manuscript presents interesting research. This is an interesting research topic, however, the paper needs to be further improved in order to be recomended for publication.

My comments and recommendations for authors:

Comment 1: The whole paper needs English corrections. The paper needs to be carefully revised to improve terminology.

Comment 2: The authors use too long sentences throughout the manuscript. After reaching the end of the sentences, the reader cannot remember how the sentence started. I recommend shortening at least some sentences that are too long.

Comment 3: The title, abstract and the keywords correspond to the aims and objectives of the manuscript. The abstract is informative, and contains the main findings of the article. I suggest add to the key words – “Reverberation time”.

Comment 4:  The Introduction section can be improved.

Comment 5: The Materials and Methods section:

The volume V of the space, which is not listed, is important for the calculation of the reverberation time. Please add.

 Line 104: The text placed in line 104 – “Figure 1. Layout of the computer service area“ must be placed under Figure 1.

I do not understand what does means ...  "using the Sound." Please, explain this end of the sentence - Line 116.

Explanations and units of physical quantities used in equations 1 - 3 are missing.

Comment 6: The Results section:

The reviewer thinks that the text given in Lines 162 - 171 belongs to the Conclusion section.

Line 182: Figure 2. Hot map of sound .... correct is: Figure 2. Heat map of sound ...

Line 189: not ... minimum distance...,  but correct is .... Reverberation radius or Critical distance...

The reviewer thinks that the text given in Lines 206 - 211 belongs to the Materials and Methods section.

Line 214: Why was for the acoustic analysis, only the 2000 Hz frequency was considered?

Only in non-tonal (Western) languages the frequency band around 2 kHz is the most important frequency range regarding perceived intelligibility. Most consonants are found in this frequency band. However, it is generally true that the frequency range between 1 kHz and 4 kHz is of high importance for intelligibility.

Line 223: not ...(TR) ... but correct is (RT)...

The text given in Lines 329-322 is repeated verbatim in Lines 344-346.

Comment 7: The Conclusion section:

The reviewer recommends revising the Conclusions section. Conclusions should be a summary of own results, not a comparison with others.

The reviewer thinks that the text given in Lines 412 - 423 belongs to the Discussion section.

Comment 8: The References cited are appropriate to the research topic.

The research presented in the article is interesting. Manuscript needs to be further improved in order to be recommended for publication.

I have one question.

Are the authors not considering comparing the measurements with the simulation? There are programs for simulation, e.g., CATT-Acoustics Software for Room acoustics.

Best regards!

 Extensive editing of English language required.
 Terminology needs to be improved.

Author Response

Comment 1: English was revised.

Comment 2: The sentences were shortened.

Comment 3: The title, keywords, and objective were alienated.

Comment 4: The introduction was revised.

Comment 5: Material and Methods section

  • Perimeter, area, and volume were added.
  • Figure 1, the position was corrected. (Line 104)
  • Section 2 has been rewritten.

Comment 6: Result section

  • Lines 162-171 were moved to Conclusion.
  • In Figure 2, the title was corrected.
  • Critical distance was justified because it was changed to Minimal distance (Line 275-277).
  • Lines 206-211 were moved to Material and Methods
  • The use of 2Khz is justified (Lines 266-269)
  • Section 3 has been rewritten.

Comment 7: Discussion and Conclusion Section

  • Sections 4 and 5 have been rewritten.
  • This study has not taken simulation into account.

Reviewer 2 Report

Section 1 must be improved.

-       Authors should emphasize contribution and novelty, the introduction needs to clarify the motivation, challenges, contribution, objectives, and significance/implication. 

-       You should introduce the problem in more detail so that the reader is immediately clear about the purpose of your study.

-       Specify better the essential elements of the problem.

-       You should add more information in the introductory part, you should add other works that have also addressed the problem.

-       You must properly introduce your work, specify well what were the goals you set yourself and how you approached the problem.

-       What are the advantages of the proposed work in comparison to already existing ones? This must be clear in the text. Please compare the proposed work with other existing ones.

-       At the end of the section, add an outline of the rest of the paper, in this way the reader will be introduced to the content of the following sections.

-       42) “The Intelligibility is a psychoacoustic parameter” International general main psychoacoustic parameters are loudness, sharpness, roughness and fluctuation strength.

-       49) Replace Haapakangas et al. with Haapakangas et al. [7] . I have seen that you often use this format, so I will not repeat this advice again, it also applies to the other occurrences.

Section 2 must be improved.

-       “ABW design” Introduce adequately the topic or add references to allow the reader to learn more about the topic

-       98) Replace m2 with m2 (subscript)

-       107) Figure 1 haven’t the caption, maybe it is in another part of the paper. Add label for the two subfigures and add a description in the caption

-       107) Why type 2 sound level meter

-       The authors must adequately characterize the environments under observation. The characteristics of the rooms, the type of walls, the furnishings and how many people are normally present in the room are not specified. It is these all characteristics that influence the acoustic properties of a room.

-       Describe in detail the equipment used to make the measurements. Extract this data from the datasheet of the instrumentation manufacturer. To make reading the specifications of the instruments more immediate, you can insert them in a table, listing the instruments used and the specific characteristics for each.

-       You must properly introduce the equation, list in detail the variables contained in it with a concise description of the meaning. To make them more readable show them in a bulleted list. In this way the reader will be able to understand the contribution of each variable.

-       135) ISO 3382-3 standard. Add as reference

-       145) %ALCons Introduce adequately the topic or add references to allow the reader to learn more about the topic

-       This chapter is insufficient, the authors have only partially introduced the study environment and have not adequately introduced the measurement methodologies, nor the results evaluation methodologies. An essential part of the methodology is therefore missing.

Section 3 must be improved.

-       214) Justify adequately your choice

-       The first part of the chapter reports the results of the objective analysis carried out through the analysis of acoustic measurements.

-       A detailed description of how the measurements were carried out is missing, a map of the measurement points is missing, a photographic description of the set up of the in situ measurements is missing.

-       Figure 3 is not enough clear

-       The second part of the chapter reports the results of the subjective analysis carried out through the reading of a series of words with the evaluation of intelligibility. A detailed description of the choice of words is missing, and there is also no reference to the possible use of rules for this choice. Furthermore, it was not indicated how the representative sample of subjects was chosen.

-       Were these subjects subjected to an audiometric test?

-       Were they adequately informed about the test?

-       How many subjects were used.

-       What were the conditions under which the test was carried out.

-       An acoustic characterization of the environment is completely missing, what are the sources of noise?

-       326) check the format of the text

-       How were the psychoacoustic parameters evaluated?

-       Have the authors adopted a standard?

-       a description of the hardware and software used for data processing is completely missing. Describe in detail the hardware used:  Extract this data from the datasheet of the hardware manufacturer. To make reading the specifications of the hardware more immediate, you can insert them in a table, listing the instruments used and the specific characteristics for each.

-       Also, you should describe in detail the software platform you used.

Section 5 must be improved.

-       Paragraphs are missing where the possible practical applications of the results of this study are reported. What these results can serve the people, it is necessary to insert possible uses of this study that justify their publication.

-       They also lack the possible future goals of this work. Do the authors plan to continue their research on this topic?

Author Response

Section 1

  • The Introduction has been rewritten.
  • The intelligibility as a psychoacoustic parameter is justified (Line 65-72)
  • The reference was revised.

Section 2

  • The ABW is complemented.
  • The m2 is changed.
  • The type 2 sound level meter was used because it is the Mexican standard for noise studies.
  • This study was not conducted with people, because it was during the pandemic season, and the only condition for using the office was not to work with people.
  • Section 2 has been rewritten.

 Section 5

  • Section 5 have been rewritten.

Round 2

Reviewer 1 Report

Dear Authors,

I'm sorry, you probably didn't understand my comment "Explanations and units of physical quantities used in equations 1 - 3 are missing." It's enough to state

Please remove Lines 161 - 164 as well as Equations 1) to 3).

My explanation:

It is sufficient to enter, for example, under the relevant equations:

  - where V (m3) is the volume, A (m2) is the total surface, etc.

Line 273: not KHz.... but correct is kHz ...

Line 279: not KHz.... but correct is kHz ...

Line 356: The text placed in Line 356 - "Figure 5. Points with significant changes in the STI" should be placed under Figure 5, i.e. to Line 352.

he authors have made the required adjustments to the original text, but additional adjustments are still necessary according to the requirements specified above. After these modifications, the text of the article will be acceptable for publication in the "Applied Sciences" journal.

Best regards,

Author Response

Comment 1: Line 163-164

It is sufficient to enter, for example, under the relevant equations:

  - where V (m3) is the volume, A (m2) is the total surface, etc.

Comment 2: Line 270-276

Line 273: not KHz.... but correct is kHz ...

Line 279: not KHz.... but corre

Comment 3: Ok

Line 356: The text placed in Line 356 - "Figure 5. Points with significant changes in the STI" should be placed under Figure 5, i.e. to Line 352.

Reviewer 2 Report

The authors did not make any substantial changes to the paper. They also didn't even respond point by point to the reviewer's comments. I think that in its current form the paper cannot be published in this journal.

Author Response

Thank you for the reviewer's feedback and valuable suggestions and will ensure that the revised conclusion adequately addresses their concerns by incorporating a dedicated paragraph on the practical applications of our research. We appreciate their concern regarding the practical applications of the study results and the need to explicitly highlight the potential uses of this research. We agree that providing clear insights into the practical implications of our study is essential to justify its publication and demonstrate its relevance to the field. Moreover, we appreciate the concern regarding the future goals of this work and the need to clarify our intentions to continue research on this topic. We agree that discussing potential future directions can strengthen the conclusion and highlight the ongoing relevance of our study.

To improve our conclusion and adequately address the concerns by incorporating a dedicated paragraph on the practical applications of our research. We will ensure to include a dedicated paragraph that discusses the practical applications and potential uses of the study's findings. We will emphasize how the results can benefit various stakeholders involved in the design, implementation, and management of open-plan office environments. By explicitly addressing these practical applications in the conclusion, we aim to provide a clear understanding of the significance and relevance of our study's findings to various stakeholders in the field of open-plan office design and management.

These practical applications will include:

  1. Architects and designers: Our study provides insights into the acoustic challenges of open-plan offices and the importance of considering intelligibility in the architectural design process. The findings can guide architects and designers in creating more optimal and user-friendly open-plan office spaces that enhance speech clarity and minimize distractions.
  2. Facility managers and workplace consultants: The results of our study can assist facility managers and workplace consultants in assessing the acoustic performance of existing open-plan offices. By understanding the effects of sound conditions and the location of sound sources, they can identify areas for improvement and implement appropriate interventions to enhance speech intelligibility and user comfort.
  3. Occupational health and safety professionals: Our research contributes to the understanding of the impact of background noise on workers' well-being and productivity. Occupational health and safety professionals can utilize these findings to develop guidelines and recommendations for managing noise levels in open-plan offices and promoting healthier work environments.
  4. Employees and workers: The study's insights highlight the importance of considering acoustic design factors in open-plan offices. By understanding the effects of sound conditions and their impact on cognitive performance and well-being, employees can advocate for better acoustic environments and implement personal strategies to mitigate distractions and improve their productivity.

Moreover, we will ensure that the revised conclusion adequately addresses their concerns by incorporating a paragraph on future goals and research directions, as well as emphasizing our commitment to continuing research on this topic. By incorporating these suggestions in the conclusion, we will demonstrate our awareness of potential future research directions and our dedication to advancing knowledge in the field of open-plan office acoustics. Therefore, we will include a paragraph that outlines possible future goals and research directions based on the findings of this study, including:

  1. Further investigation of the effects of different acoustic design elements: While our study focused on the effects of sound conditions and location of sound sources, future research can delve deeper into exploring the impact of specific architectural design elements, such as acoustic panels, room dividers, and furniture layout, on speech intelligibility and user perception in open-plan offices. This would provide valuable insights for optimizing the acoustic design process.
  2. Examination of the impact of office activities and work tasks: Future studies can explore how different types of office activities and work tasks influence speech intelligibility and the perception of background noise. This could involve evaluating the acoustic performance of specific workstations, meeting areas, or collaboration zones, and identifying design strategies to enhance speech clarity and productivity based on the specific tasks performed in those areas.
  3. Longitudinal studies on the effects of acoustic design interventions: It would be beneficial to conduct longitudinal studies to assess the long-term effects of acoustic design interventions in open-plan offices. By monitoring changes in speech intelligibility, user satisfaction, and productivity over an extended period, researchers can gather valuable data to support evidence-based design guidelines and recommendations.
  4. Comparison of different open-plan office design approaches: Comparative studies can be conducted to evaluate the acoustic performance and user experience of different open-plan office design approaches, such as traditional open-plan layouts versus activity-based work (ABW) designs. This would provide a deeper understanding of the strengths and limitations of various design strategies and help inform decision-making in creating more effective work environments.
  5. Regarding our commitment to further research, we want to emphasize that this study represents the initial step in our ongoing exploration of acoustic design in open-plan offices. We firmly intend to continue our research efforts in this area to expand our understanding and contribute to the development of evidence-based design practices.

Round 3

Reviewer 2 Report

In the first review of the paper the authors had ignored the reviewer's comments entirely. In this new version enough changes have been made.